# Effects of Ni Content on Energy Density, Capacity Fade and Heat Generation in Li[Ni_x_Mn_y_Co_z_]O_2_/Graphite Lithium-Ion Batteries

**DOI:** 10.3390/mi16101075

**Published:** 2025-09-23

**Authors:** Gaoyong Zhang, Shuhuang Tan, Chengqi Sun, Kun Zhang, Banglin Deng, Cheng Liao

**Affiliations:** 1Naval Architecture and Shipping College, Guangdong Ocean University, Zhanjiang 524088, China; zhanggaoyong27@gmail.com (G.Z.); 11821421ff1@stu.gdou.edu.cn (S.T.); kunzhang@gdou.edu.cn (K.Z.); dengbanglin123@126.com (B.D.); 2BYD Industries Co., Ltd., Shenzhen 518118, China; chenglhnu@163.com

**Keywords:** lithium-ion battery, Li[Ni_x_Mn_y_Co_z_]O_2_, electrical performance, capacity fade, heat generation

## Abstract

The demand for high energy density in mobile devices (including vehicles and small ships) is increasing. Nickel–Manganese–Cobalt (NMC) ternary, as a battery cathode material, is increasingly being applied because of its higher energy density relative to LiFePO_4_ or other traditional materials. But NMC also faces challenges, such as a high degeneration rate and heat generation. So these aspects of Ni content must be clarified. In the current study, two Ni-content battery cells were tested, and the results of other composition cathode cells from the literature were compared. And three typical Ni-content batteries were simulated for searching Ni effects on performance, capacity fade and heat generation. Some findings were achieved: (1) from 0.8 Ni content, it can be seen that the specific capacity growth rate (slope) was much greater than before; (2) cathode materials that have an odd number (that does not surpass 0.7) of Ni content showed a linear capacity degradation trend, but others did not; (3) the Li concentration within material particles did not correspond to absolute stress value but stress temporal gradient; and (4) during discharge, lower Ni content made the heat peak occur earlier but lowered the absolute value; the irreversible heat increased with Ni content non-linearly, so that the higher the Ni content went up, the higher the increase rate of the irreversible heat ratio. Thus, the results of this study can guide the design and application of high energy batteries for mobile devices.

## 1. Introduction

As the global energy crisis becomes increasingly severe [1], countries around the world are actively advancing decarbonization strategies and promoting the adoption of green and sustainable energy sources [2]. A key bottleneck in the global shift toward net-zero emissions and renewable energy adoption is the efficient storage of electrical energy, primarily relying on battery technologies [3,4]. From powering energy storage systems in electric vehicles [5] to storing surplus electricity generated during peak production in renewable-integrated power systems for grid load balancing [6], and further to energy storage applications in the maritime sector, particularly in modern vessels equipped with advanced gas turbine and electric propulsion systems [7,8], battery technologies have demonstrated their critical role in advancing the development of sustainable energy systems. It is precisely the growing technological demands across various fields that have propelled the rapid advancement of battery storage technologies.

Among various types of energy storage systems, lithium-ion batteries (LIBs) have stood out due to their outstanding performance characteristics, including high energy density, excellent energy conversion efficiency, lightweight design and having a low cost and long cycle life, attracting widespread attention and extensive research [9]. Currently, the development of LIBs technology faces two primary challenges: one is the further improvement of electrochemical performance, and the other is the assurance of cycle life and operational safety [10]. Since 2010, the number of vessels using LIBs as Battery Energy Storage Systems (BESS) has increased significantly. By 2024, approximately 1200 fully electric or hybrid ships had already entered operation or were under design and construction [11]. Practical applications have demonstrated that LIBs, as a power source for ships, are applicable to nearshore navigation within 100 nautical miles, while long-distance ocean-going voyages must rely on hybrid propulsion systems [12,13]; this limitation is imposed by the performance of LIBs. Bugryniec et al. emphasized the risks associated with the thermal runaway (TR) in shipboard BESS and conducted a retrospective analysis of past maritime BESS incidents [14]. Another study, conducted by Wu et al., systematically discussed the risks of lithium-ion batteries in maritime transport and reviewed the corresponding methods for risk identification and assessment [15]. The safety issues associated with lithium-ion batteries (LIBs) have become increasingly difficult to ignore. In July 2025, a cargo ship transporting electric vehicles from China to Mexico was engulfed in a severe fire caused by a lithium battery thermal runaway, ultimately resulting in the vessel’s sinking [16].

The cathode material plays a decisive role in determining the battery’s voltage and capacity, and also serves as a limiting factor for lithium-ion transport kinetics and cycle durability [17]. Therefore, in-depth investigation and analysis of cathode materials represent a key pathway toward improving lithium battery performance, safety and life cycle. LiCoO_2_ is one of the most classic and earliest commercialized cathode materials, featuring a very high energy density and decent cycle life, but it has increasingly been criticized for its high cost and poor thermal stability. Moreover, after multiple generations of development, its energy density has approached the theoretical limit, leaving limited room for further improvement or practical advancement [18]. While olivine-structured LiFePO_4_ exhibits a lower energy density compared to LiCoO_2_, it demonstrates outstanding thermal stability and an excellent cycle life, along with a lower cost, making it a dependable choice for stationary energy storage and electric vehicle applications [19]. Spinel-structured LiMn_2_O_4_ has attracted considerable attention due to its low cost, environmental friendliness, high voltage plateau near 4.0 V, and excellent rate capability enabled by its three-dimensional lithium-ion diffusion pathways. However, its cycling stability is limited by manganese dissolution and structural degradation at elevated temperatures [20].

Among various cathode chemistries, layered nickel–manganese–cobalt oxides Li(Ni_x_Mn_y_Co_z_)O_2_ (commonly referred to as NMC) have emerged as a leading candidate for next-generation lithium-ion batteries due to their high specific capacity, tunable metal ratios and strong commercial viability. Ni-rich NMCs offer higher theoretical energy density and improved electrochemical performance compared to LiCoO_2_, while achieving a cycle life approaching that of LiFePO_4_. In appropriately engineered systems, they also exhibit enhanced thermal stability and safety relative to both LiCoO_2_ and LiMn_2_O_4_. Furthermore, NMC materials possess inherent cost advantages, particularly in compositions with reduced cobalt content. One of the primary research strategies for NMC materials involves adjusting the ratios of transition metals (TMs)—particularly by increasing the nickel content—to enhance the electrochemical performance of the battery while reducing reliance on cobalt, a costly and geopolitically sensitive element [21]. High-nickel NMC compositions, such as NMC0.89-0.055-0.055 and NMC811, exhibit superior electrochemical properties, including higher specific capacity and energy density, and they have thus been widely adopted in modern electric vehicle platforms [22]. However, increasing the nickel content introduces several challenges, including structural instability, aggravated cation mixing, accelerated capacity degradation and heightened thermal reactivity. These issues pose significant risks to battery safety and long-term reliability, especially under elevated temperatures or aggressive cycling conditions [23]. To address these concerns, extensive research has investigated the electrochemical behavior of NMC materials with various TM ratios and has identified critical degradation mechanisms such as microcrack formation, phase transitions, oxygen release and surface deterioration. Among these NMC materials, NMC111 is well-regarded for its robust lattice structure and excellent thermal stability under prolonged cycling. Its relatively low nickel content helps minimize lattice distortion and preserve structural integrity during repeated charge–discharge processes, though it also results in a limited reversible capacity [24]. NMC622 offers significantly higher energy density than NMC111 and, unlike NMC811, which is prone to severe oxygen release and intergranular cracking under high-voltage operation, it exhibits substantially reduced lattice oxygen loss, thereby enhancing cycling reliability and lowering thermal risk. As such, it is frequently used as a reference material in both academic research and industrial practice [25]. NMC712 has been regarded as a representative composition in the exploratory search for balanced TM ratios between medium- and high-nickel regimes. It aims to achieve a reasonable trade-off between electrochemical performance and structural robustness. Theoretically, NMC712 offers better thermal and structural stability than NMC811, while still delivering high energy density [26].

Although extensive research has been conducted on NMC cathode materials with varying nickel content with respect to how much Ni content is chosen and how to design and applied the NMC batteries, the community still needs a systematic investigation. In the current study, two Ni content cells, NMC811 and NMC712, were tested, and data of other Ni content batteries were added from the literature. The electrochemical performance, capacity fading (cycle life) and mechanical stress–strain response (structural stability) and thermal behavior (safety) of these materials were systematically analyzed. We hope to clarify the specific impact of nickel content in NMC materials on lithium-ion battery behavior and provide new insights for the design and optimization of NMC-based cathode materials.

The remainder of this paper is organized as follows: Section 2 presents the experimental methodology, and Section 3 outlines the COMSOL-based multiphysics modeling framework that is employed to simulate mechanical stress–strain evolution and heat generation across different NMC compositions. Section 4 presents and discusses the experimental and simulated results, including the specific capacity, capacity degradation and thermal reactivity of various cathode materials. Finally, Section 5 summarizes the key findings and recommendations for future research.

## 2. Experiments

Two pouch-type lithium-ion cells were procured directly from Farasis Energy Company (Ganzhou, Jiangxi Province, China) for testing. One cell employed a Li[Ni_0.8_Mn_0.1_Co_0.1_]O_2_/graphite (NMC811) configuration, while the other used a Li[Ni_0.7_Mn_0.1_Co_0.2_]O_2_/graphite (NMC712) configuration. In both cells, the electrolyte consisted of LiPF_6_ (approximately 13 wt%) dissolved in a mixture of ethylene carbonate and ethyl methyl carbonate (EC/EMC). The nominal capacities were 78 Ah for the NMC811 cell and 75 Ah for the NMC712 cell, with a maximum operating voltage of ~4.25 V for both. The two cells shared identical dimensions of 534 mm (length) × 103 mm (width) × 8.2 mm (thickness). The test equipment and corresponding technical parameters are summarized in Table 1.

The experimental procedures have been described in detail in our previous work [27]. Therefore, only a concise summary is provided here. The experimental system layout and data flow are illustrated in Figure 1. All cells were placed in a thermostatic chamber capable of maintaining a constant ambient temperature within the range of −40 to 85 °C. During the cycling tests, the NMC811 cell was operated at 25 °C with a 1C current rate, whereas the NMC712 cell was tested at 45 °C with a 1/3C current rate. A programmable charge/discharge system was employed to conduct the cycling under predefined protocols, as detailed in Table 2. The electrochemical performance was evaluated initially and subsequently after every 100 cycles.

## 3. Simulations

In this study, all simulations were carried out using the software COMSOL Multiphysics version 6.2. As illustrated in Figure 2, considering the layered architecture of the Li-ion battery pouch cell, a one-dimensional (1D) geometry was adopted to represent the pseudo-two-dimensional (P2D) battery model [28]. This model assumes: (1) the electrode material is composed of spherical particles; (2) the double layer electric effect is ignored; and (3) the conductivity of the current collectors is very high, so electrochemical reaction kinetics only take effect on the *x*-axis. The thickness of the anode, separator and cathode is 63 μm, 12 μm and 46 μm, respectively.

The P2D model describes the Li-ion transformation in the electrolyte and inside the active material particles. Within the electrolyte, both diffusion (dependent on concentration gradient and liquid diffusion coefficient) and migration (dependent on potential distribution in the liquid phase) are considered—see Equation (1); while for the latter, the second Fick’s Law is used to describe the solid-phase diffusion of lithium ions, as described by Equations (2) and (3). The charge transfer that occurs at the interface between the electrolyte and the electrode particles is solved by the Butler–Volmer equation, as shown in Equations (4) and (5); thus the thermal phenomenon is coupled (through Equation (6)), and details can be seen in Refs. [29,30].(1)$$N_{i}=-D_{i}\nabla c_{i}-z_{i}u_{m,i}Fc_{i}\nabla \phi_{l}+c_{i}u$$(2)$$\frac{\partial c_{1,p}}{\partial t}=\frac{1}{r_{p}^{2}}\frac{\partial }{\partial r_{p}}\left(r_{p}^{2}D_{1,p}\frac{\partial c_{1,p}}{\partial r_{p}}\right)$$(3)$$\frac{\partial c_{1,n}}{\partial t}=\frac{1}{r_{c}^{2}}\frac{\partial }{\partial r_{c}}\left(r_{c}^{2}D_{1,n}\frac{\partial c_{1,n}}{\partial r_{c}}\right)$$
In the above equations, *N_i_* is the total flux of species *i* (mol/(m^2^·s)), *D_i_* is the diffusion coefficient (m^2^/s), *c_i_* represents the concentration of the ion *i* (mol/m^3^), *z_i_* is the valence, *u*_*m*,*i*_ is mobility (s·mol/kg), *F* denotes the Faraday constant (C/mol), *φ_l_* is electrolyte potential, u is the velocity vector (m/s); *c*_*1*,*p*_ and *c*_*1*,*n*_ are, respectively, the Li concentration in positive and negative solid particles (mol/m^3^), *r_p_* and *r_c_* (Figure 2) are, respectively, the positive and negative particle diameters (m) and *D*_*1*,*p*_ and *D*_*1*,*n*_ are the diffusion coefficients (m^2^/s), respectively, of positive and negative particles.(4)$$i_{n}=i_{0,n}\left[\exp\left(\frac{\alpha_{a}F\eta }{RT}\right)-\exp\left(\frac{\alpha_{c}F\eta }{RT}\right)\right]$$(5)$$i_{0,n}=Fkc_{s,surf}^{\alpha_{c}}c_{l}^{\alpha_{a}}{\left(c_{s,\max}-c_{s,surf}\right)}^{\alpha_{a}}$$
where *α_a_* and *α_c_* are the charge transfer coefficients, respectively, of the anode and cathode, *η* is the electrode overpotential (V), *R* represents the gas constant (J/mol·K), *T* is the temperature (K), *k* is the reaction rate constant (m^2.5^/mol^0.5^·s), *c*_*s*,*surf*_ and *c*_*s*,*max*_ denote, respectively, the Li-ion concentration at the particle surface and the maximum concentration (mol/m^3^) and *c_l_* is the electrolyte concentration (mol/m^3^).(6)$$\rho_{i}C_{p,i}\frac{\partial T}{\partial t}+\nabla \left(-\lambda \nabla T\right)=Q_{act}+Q_{rea}+Q_{ohm}$$
where *ρ_i_*, *C*_*p*,*i*_, and *λ* are density (kg/m^3^), specific heat at constant pressure (J/kg·K) and heat conductivity (W/m·K), respectively. *Q_act_*, *Q_rea_* and *Q_ohm_* denote, respectively, the polarization heat, the reaction heat and the ohmic heat (J).

Lastly, Li insertion and extraction make active material’s volumetric changes, thus inducing mechanical stress, as described by Equations (7) and (8) in particle scale and Equation (9) in electrode scale. This phenomenon is one of the most important reasons that cause battery capacity degeneration. So, we calculated the stress to indirectly analyze the degeneration under different cathode Ni content materials. Three typical Ni content cathode materials, namely, NMC811 (LiNi_0.8_Mn_0.1_Co_0.1_O_2_), NMC622 (LiNi_0.6_Mn_0.2_Co_0.2_O_2_) and NMC532 (LiNi_0.5_Mn_0.3_Co_0.2_O_2_), were comparatively simulated with an invariable anode or electrolyte such that the effect of Ni-content can be focused. Although these two parts also have a certain degree of impact on battery performance [31], all the data sources in the current study used a graphite anode and similar electrolytes, so the assumption of an “invariable” anode and electrolyte is reasonable. To evaluate the stress–strain response, the volumetric change of active material particles during Li insertion and extraction must be predefined. The volumetric change data for the three NMC cathode materials and the anode material are presented in Figure 3. As observed, the relative volumetric change increases with higher Ni content, while the anode exhibits the largest volumetric change overall. In addition, as shown in Figure 2, the homogeneous material diameter was assumed, the mean anode diameter *r_c_* is 5 μm and the mean cathode diameter *r_p_* is 6 μm. According to Ref. [32], the Young’s Moduli is 78 GPa for both NMC811 and NMC622 and 90 GPa for NMC532. Finally, all simulated cases were under a 1C current rate.(7)$$\varepsilon_{rr}^{*}=\frac{1}{E_{p}}\left(\sigma_{rr}^{*}-2\upsilon_{p}\sigma_{\theta \theta }^{*}\right)+\frac{\Omega }{3}∆c+\alpha ∆T$$(8)$$\varepsilon_{\theta \theta }^{*}=\frac{1}{E_{p}}\left(\sigma_{\theta \theta }^{*}-\upsilon_{p}(\sigma_{\theta \theta }^{*}+\sigma_{rr}^{*})\right)+\frac{\Omega }{3}∆c+\alpha ∆T$$(9)$$\varepsilon =\frac{1}{E}\left((1+\upsilon )\sum_{ij}-\upsilon_{p}\sum_{kk}\delta_{ij}\right)+\frac{\Omega }{3}∆c\delta_{ij}+\alpha ∆T\delta_{ij}$$
where *ε_rr_* and *ε_ϴϴ_* are the radial and tangential strain, respectively. *σ_rr_* and *σ_ϴϴ_* are the radial and tangential stress (Pa), respectively. *E* and *υ* Young’s modulus (Pa) and Poisson’s ratio, respectively, and *p* denotes particle scale. Ω is partial molar volume (m^3^/mol). Δ*c* is the difference of the Li-concentration with the reference concentration (mol/m^3^). α is the coefficient of thermal expansion (1/K). *Σ_ij_* and *Σ_kk_* are the stress in electrode level and particle level (Pa), respectively. *δ_ij_* is the Dirac delta function.

## 4. Results and Discussion

### 4.1. Electrical Performance

The specific capacity of lithium-ion battery cathode materials is a key indicator for evaluating their energy storage potential. As shown in Figure 4, the currrent test shows that NMC712 and NMC811 cathode materials exhibited initial reversible specific capacities of 160–195 mAh g^−1^ (the result was corrected from 1/3C to 1C) and 175–205 mAh g^−1^, respectively. To place these results in a broader context, the following data from the literature were also included: LiFePO_4_ typically shows a specific capacity at 145–163 mAh g^−1^ [33], NMC111 at 140–170 mAh g^−1^ [34], NMC532 at 155–186 mAh g^−1^ [35], NMC622 at 168–185 mAh g^−1^ [25] and NMC0.89/0.055/0.055 (denoted as NMC89) at 195–238 mAh g^−1^ [22]. But interestingly, pure LiNiO_2_ (LNO) is less applied as a cathode material; although its theoretical capacity is 275 mAh g^−1^, its practical capacity is lower even than NMC111 [36] due to Li^+^/Ni^2+^ cation mixing and Jahn Teller distortion of the Ni-O octahedron, and thus the cycling and stability performances are poor [37]. So LNO is ignored in the current study.

A key finding of this study is the clear positive correlation between nickel content and specific capacity among the NMC materials examined. Systematic comparison reveals a direct trend: higher Ni content leads to higher capacity. This can be attributed to the enhanced contribution of the Ni^2+^/Ni^4+^ redox couple, which dominates the electrochemical activity in nickel-rich compositions. As nickel content increases, the number of active redox centers per formula unit also increases, allowing more Li^+^ ions to be inserted and extracted during each cycle, thereby raising specific capacity. This trend is particularly evident when comparing low-Ni compositions such as NMC111 and NMC532 to high-Ni variants like NMC811 and NMC89, as shown in Figure 4. From 0.8 Ni content the specific capacity growth rate (slope) is much greater than before.

In addition, Figure 4 includes a comparison between the NMC materials and the currently widely used LiFePO_4_ material. The results clearly show that Ni-rich NMC materials offer significantly higher specific capacities than LiFePO_4_, underscoring their performance advantage. While LiFePO_4_ is favored for its outstanding thermal stability and cycling life, its relatively low theoretical capacity limits its potential for further development and high-energy applications. In contrast, NMC materials possess significantly higher theoretical capacities, enabling greater energy output per unit mass, thereby better supporting the performance demands of high-energy-density application scenarios. These findings collectively confirm the decisive impact of Ni content on the improvement of specific capacity in layered oxide cathode materials and further clarify its core role as the main active component in capacity contribution.

It is noteworthy that all materials exhibit a broad range of capacity values, reflecting fluctuations arising from multiple influencing factors, which can be broadly classified into two categories. The first category pertains to electrochemical testing conditions, such as current density (C-rate), voltage window and ambient temperature. In general, lower C-rates allow more complete lithiation/delithiation processes, resulting in higher measured capacities [38]. For example, testing at 0.1C typically yields a greater reversible capacity compared to testing at 1C or higher. For NMC materials, the typical voltage range is 2.8 to 4.4 V, where expanding the voltage window—especially the upper cut-off—can increase capacity by activating additional redox reactions. Temperature is typically set at 25 °C during testing, and elevated temperatures tend to increase ionic conductivity and reduce charge transfer resistance, thereby improving specific capacity, although it may also accelerate side reactions and reduce stability.

The second category of influence originates from intrinsic material properties and processing strategies. Even for materials with identical nominal compositions, various developed or optimized versions exist. For instance, in the study by R. Wihba et al., the variation in specific capacity of NMC622 was demonstrated through different synthesis approaches [39]. A layer of LPO (layer polyolefin) coating to the cathode surface using ALD (atomic layer deposition) technology caused a slight decrease in specific capacity; it significantly mitigated the capacity fading during cycling, increased the chemical stability and extended the service life of the material [40]. Xu et al. reported that the incorporation of cationic dopants such as Al, Mg, Ti and Si into NMC811 effectively enhanced the initial specific capacity and/or improved capacity retention and structural stability [26]. Moreover, as observed by Soloy et al. [41], all nano-scale mean diameter materials were better than that of micro-scale particles, but within nano-scale particles, it was not the case that smaller is always better; the 400 nm mean particle diameter provided the highest performance.

Lastly, the discharge processes of two typical NMCs were chosen to verify the simulation. As shown in Figure 5, the simulation can almost reproduce the experimental results. Furthermore, the voltage of NMC622 drastically decreases after about one third of the discharge process relative to that of NMC811, resulting in a drop in capacity.

### 4.2. Capacity Fade

Capacity fade is a key issue in practical applications, particularly for cathode materials with high Ni content; therefore, understanding the fading mechanisms and trends is essential. As demonstrated in Figure 6, the extent of capacity degradation becomes more pronounced with increasing Ni content. The corresponding test conditions and fading rates are summarized in Table 3. Although the test protocols for cells with different Ni contents were not identical, the results indicate that the fading rate increases almost linearly with the Ni content in the cathode. But unexpectedly, the fading rate of NMC111 is comparable with that of NMC712. According to the review of Das et al. [31], the capacity degradation of NMC111 indeed is not low; moreover, among those cycle tests on NMC111, few studies exceeded 500 cycles. This also illustrates that in recent years, work on resisting capacity degradation of high-Ni battery have achieved great progress. In addition, NMC532’s capacity fading rate is the lowest, except for its low Ni-content, also very likely because of its small size (for consumer electronics) and very low initial capacity (only 240 mAh at 4.4 V). Finally, generally as Ni content increases, the cutoff voltage decreases. A closer examination of Figure 6 reveals several additional features: (1) the fading processes of NMC532 and NMC712 follow an almost linear trend, whereas the others do not; (2) the relative capacity of NMC622 and NMC811 exhibit an abrupt decline at approximately the 810th cycle, while the sudden drop for NMC89 occurs much earlier, around the 400th cycle; (3) however, before the sudden drop, the high Ni-content cathode (especially for NMC811) has relatively slow capacity fading. Interestingly, relative to the results of the LiFePO_4_ battery in Ref. [42], the fading rate of the NMC ternary cathode is not always higher. In fact, both the microstructure of the active materials and the macroscopic design of the cell exert a significant influence on the capacity-fading characteristics.

From Table 3, it can be seen that the fading rate of NMC111 is not low, and the high-Ni-content cathode is increasingly being applied, so only three common use cathode types (namely NMC532, NMC622 and NMC811) were simulated to explore the performance impact factors and mechanisms. The simulated results were used to analyze the effects of Ni content in the cathode on mechanical stress, which is a critical factor affecting battery capacity fading characteristic and durability [23]. Given the large volume of data, only representative results are presented here. As shown in Figure 7a, the stress gradients in both the current collector and the separator (locations identified in Figure 8 and Figure 9) are more pronounced during discharge than during charge. Figure 7c demonstrates that the peak values of both total heat dissipation and irreversible heat generation are higher in the discharge process compared to the charge process. Moreover as demonstrated in Figure 7b, at corresponding peak stress points, the strain energy of the anode exceeds that of the cathode. A sharp increase in anode strain energy occurs near the anode/separator interface, primarily because the volumetric change of the anode material is greater than that of the cathode (Figure 3). This observation is consistent with the findings reported in Ref. [44]. And the silicon alloy/graphite blend anode can limit the volume expansion and form a more stable SEI layer, thus prolonging the battery lifetime [45]. But the mechanical behaviors of high Si-content cells are more complex than traditional cells [46]. Nevertheless, in the subsequent analysis, only the results for the anode during discharge are presented for brevity.

Next, the effect of Ni content in the cathode on electrode stress is analyzed. For brevity, only three representative NMC materials—NMC532, NMC622 and NMC811—were simulated. Figure 8a shows the stress evolution at the anode–separator interface for three cathode materials. When the battery was discharged at 1C with a high-Ni cathode (NMC811), a sudden stress rise (~43 MPa) occurred at the initial stage (95 s). This phenomenon may partly explain the abrupt capacity drop observed during long-term cycling of high-Ni batteries. As Ni content decreased, this sudden increase of stress disappeared. Moreover, among the three cathode materials, in the middle stage of discharge the stress peak occurred earlier and with a lower magnitude for cathodes with lower Ni content. During other discharge stages, the stress remained relatively stable at the 10^6^ Pa level across all three materials. As illustrated in Figure 8b, the normalized Li concentration closely correlates with the stress evolution, confirming that Li deintercalation/intercalation is the primary cause of stress generation in active particles. Furthermore, from the particle center to the edge, the Li concentration shows only a slight decrease—particularly in NMC811, where the gradient is minimal—indicating highly efficient Li-ion transport within the particles.

Subsequently, Figure 9a presents the stress evolutions of the current collector for the three materials. Except for the initial sudden increase in separator of NMC811, the stress level in the current collector is comparable to that in the separator; this agrees with the results of Ref. [29]. By combining the analyses in Figure 9a,b, it can be observed that the Li concentration correlates not with the absolute stress value, but rather with the temporal gradient of stress. This is evidenced at 750 s, where the Li concentration reaches its maximum and the stress gradient is the steepest, although the absolute stress is not at its highest. Similarly, at 2332s for NMC532, 2481 s for NMC622 and 2600 s for NMC811, the stress continues to increase even though the Li concentration levels remain nearly identical.

Lastly, within the material particles, the stress level reaches ~10^8^ Pa for all three materials, which is higher than that in the separator or in the current collector. However, the increase with Ni content is relatively modest: from NMC532 (at 1401 s, Figure 8a) to NMC622 (at 1621 s), the maximum stress rises by only 1.72%, and from NMC622 to NMC811 (at 1800 s), it increases by 3.48%. Moreover, in contrast to the findings reported by Ref. [30], the maximum stress in the present study was observed near the particle center. These results are consistent with the results of Kondrakov et al. [24], who found that particle cracking initiates in the central region. It is worth noting that material modification strategies can effectively enhance stress resistance. For instance, wrapping active material particles with carbon nanotubes (CNTs) can improve their ability to withstand higher mechanical stress, thereby contributing to enhanced battery durability [47].

### 4.3. Heat Generation

The thermal runaway (TR) of lithium ion batteries is important because it will cause thermal failure [48] even in the case of a major safety accident [49]. Therefore, identifying the underlying heat sources is essential, particularly for high-Ni cathodes, which tend to generate more heat [50]. The evolutions of volumetric heat dissipation power at the anode–current collector interface for three NMC material cells are shown in Figure 10. Two distinct peaks are observed in the negative region (heat dissipation): the first occurs at approximately one-third of the discharge process, and the second at about two-thirds. Additionally, similar to the stress evolution, cells with a lower Ni content exhibit earlier heat peaks with smaller magnitudes. And the observed phase differences among the materials may be attributed to varying degrees of heat accumulation, arising from different current densities during discharge. Therefore, the electrochemical reaction also changes with Ni content. This can be proved in Figure 11 (the corresponding moments can be seen in Figure 10); it shows that the reaction source increases with Ni content, while it decreases drastically from the current collector side to the separator side.

For the positive area (absorbing heat from the current collector) the heat peak appears in the end-piece of the discharge. In the ending phase (about 3500 s), although the heat transfer is still high (Figure 10), the electrochemical reaction has greatly decreased, as displayed in Figure 11 (3501 s of NMC811).

Finally, the irreversible heat source is a critical parameter governing thermal failure. The irreversible heat ratios are provided in Figure 10. For NMC811 and NMC622, the peak ratios exceed 100%, most likely due to reverse heat transfer from the current collectors. As shown, from NMC532 to NMC622, the irreversible heat increases only slightly, with the average ratio rising by ~6.5 percentage points, while the current density increases by ~11%. In contrast, from NMC622 to NMC811 the irreversible heat rises substantially, with the average ratio increasing by ~27 percentage points despite the current density increasing by only ~13.5%. These results demonstrate that the irreversible heat increases with Ni content in a nonlinear manner: as the Ni fraction rises, the growth rate of the irreversible heat ratio becomes progressively higher.

## 5. Conclusions

In this study, through a combination of experiments and simulations, the effects of Ni content in cathode materials on the electrochemical performance, capacity degradation, mechanical stress and heat generation of Li-ion batteries were systematically investigated. The key findings are summarized as follows: (1) the specific capacity of ternary material larger than that of LiFePO_4_ increased with Ni content. Moreover, from 0.8 Ni content the specific capacity growth rate (slope) was much greater than before. (2) The capacity fading rate of the ternary material increased with Ni content; relative to the results of the LiFePO_4_ battery, the fading rate of the NMC ternary cathode is not always higher; moreover, cathode materials that have an odd number (do not surpass 0.7) of Ni content showed a linear capacity degradation trend, but others did not. (3) The Li concentration within material particles did not correspond to the absolute stress value but corresponded to the stress temporal gradient. (4) During discharge, like in stress history, lower Ni content made heat peak occur earlier with a lower absolute value. Finally, the irreversible heat increased with Ni content non-linearly; the higher the Ni content rises, the higher the increase rate of the irreversible heat ratio.

Overall, these findings provide valuable insights into the performance and safety characteristics of high-Ni cathode materials and may guide the development of Ni-rich cathode batteries in support of the ongoing transition toward green and sustainable energy systems.

## Figures and Tables

**Figure 1 micromachines-16-01075-f001:**
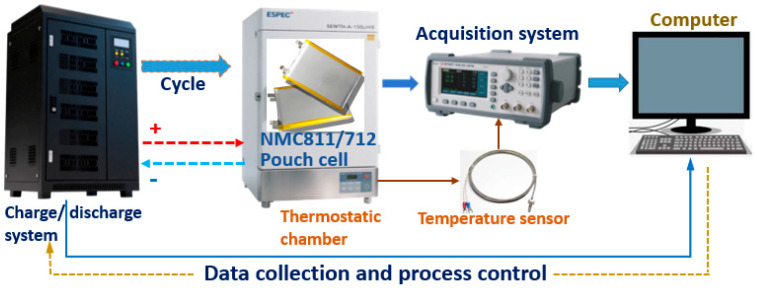
The experimental system layout and data flow.

**Figure 2 micromachines-16-01075-f002:**
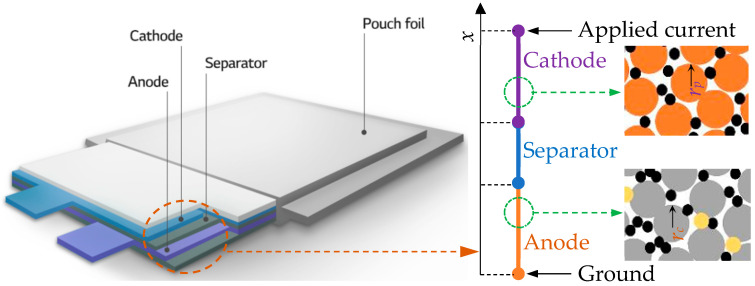
The pouch cell structure and simulated P2D model set-up.

**Figure 3 micromachines-16-01075-f003:**
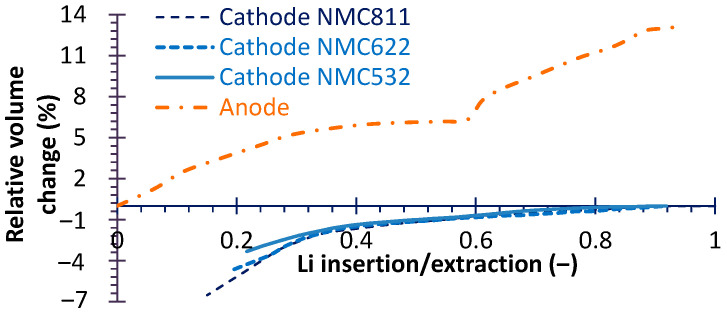
The volumetric change during Li insertion/extraction for various NMC and anode materials.

**Figure 4 micromachines-16-01075-f004:**
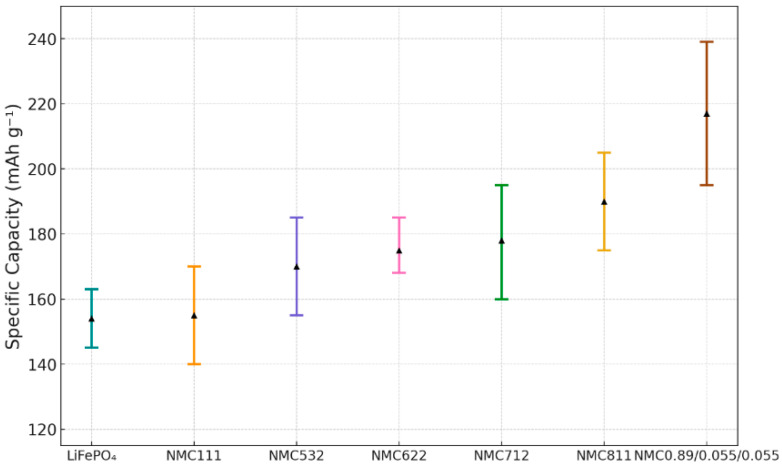
The comparisons of specific capacity for seven cathode compositions. NMC712 and NMC811 are from this study; the remaining data were compiled from the literature reports.

**Figure 5 micromachines-16-01075-f005:**
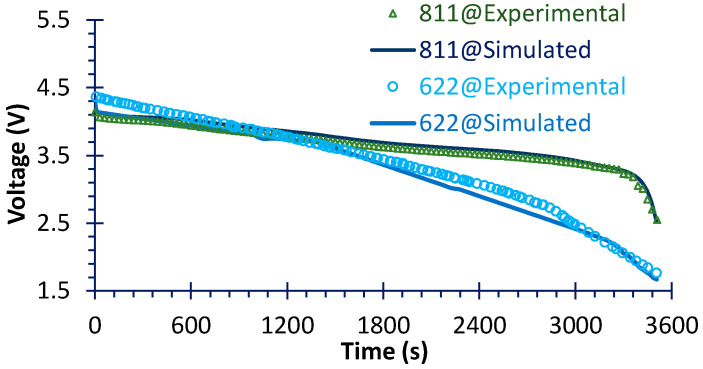
The comparisons of 1C charge performances between simulated and experimental data; the 811 data is tested by the current study and the 622 data is derived from Ref. [39].

**Figure 6 micromachines-16-01075-f006:**
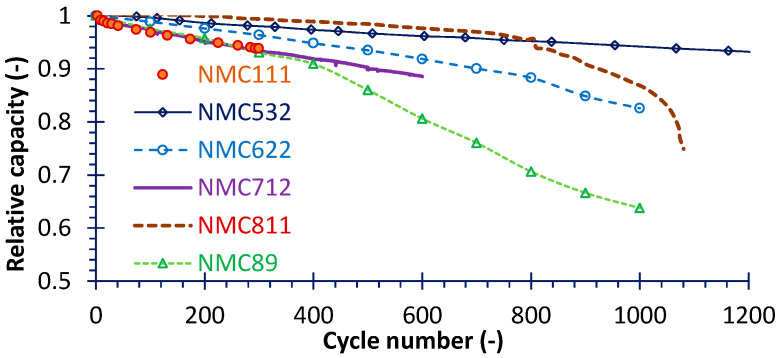
The comparisons of capacity fading for various NMC cathode material batteries; the test conditions and data sources are given in Table 3.

**Figure 7 micromachines-16-01075-f007:**
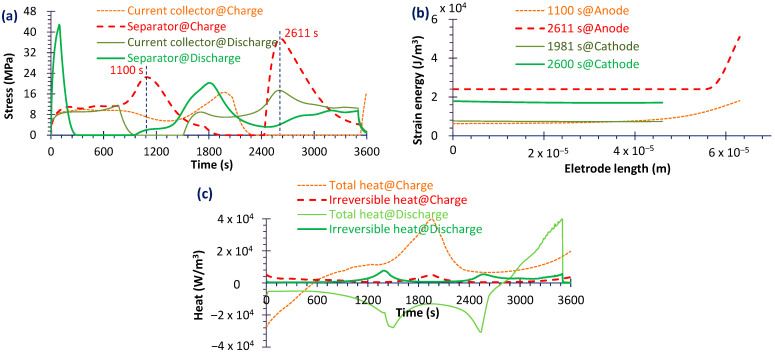
Stress comparison during charge and discharge (**a**), strain energy comparison between anode and cathode at some specific moments (**b**) and heat comparison during charge and discharge (**c**); all results are simulated on NMC811.

**Figure 8 micromachines-16-01075-f008:**
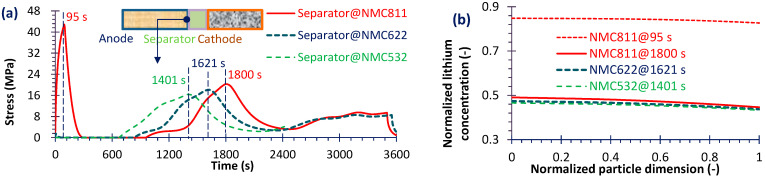
Separator stress history during discharge (**a**) and normalized Li concentration through anode material particle (0 represents the particle center) at some stress peak moments (**b**).

**Figure 9 micromachines-16-01075-f009:**
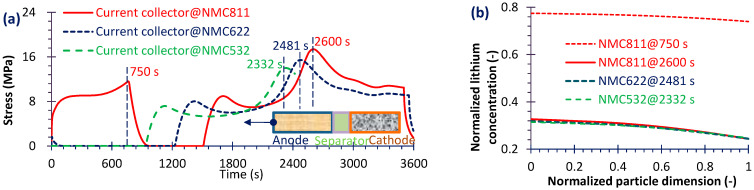
Current collector (negative) stress history during discharge (**a**) and normalized Li concentration through anode material particle at some stress peak moments (**b**).

**Figure 10 micromachines-16-01075-f010:**
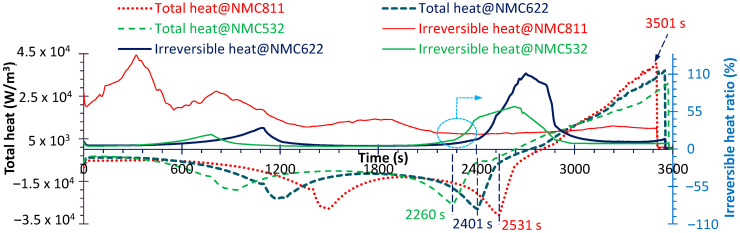
Total volumetric heat dissipation power (left y-axis) and irreversible heat ratio (right y-axis).

**Figure 11 micromachines-16-01075-f011:**
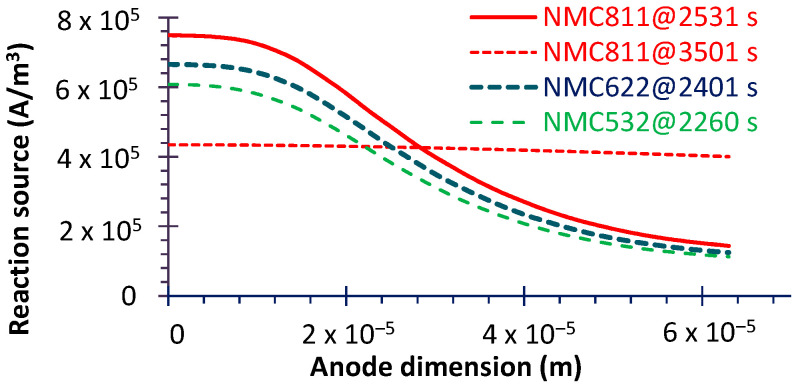
Electrode (anode) reaction source at some specific moments (indicated in Figure 10).

**Table 1 micromachines-16-01075-t001:** Technical parameters of the main experimental equipment.

Equipment	Brand and Model	Range	Accuracy	Error
Thermostat	ESPEC SEWTH-A-150LHS (ESPEC Corp., Fukuchiyama, Japan)	−40–85 °C	±0.3 °C ±2.5% RH	±1.0%
Charge/Discharge System	RUINENG-HRCDS-100V (Ruineng Pumps (Fujian) Co., Ltd., Quanzhou, China)	0–100 V 0–200 A	±0.05% F.S ±0.02% V/A	±0.1%
Data Collector	KEYSIGHT-34970A (Keysight Technologies (formerly Agilent), Loveland, CO, USA)	DC/AC	0.0035%	/

**Table 2 micromachines-16-01075-t002:** Cell cycle process (steps).

No.	Procedure	Step Details	Notes
1	Parameter calibration	Check voltage/current range, accuracy (±0.05% F.S. for voltage and ±0.02% V/A for current) and system connection. Calibrate temp sensor (±1 °C max error)	Record and store verification data
2	Set ambient temp: 25/45 °C (811/712)	Maintain temp within thermostat for 30 min, keep fluctuation ≤ ±0.5 °C	/
3	Charge to 100% SOC under 1C or 1/3C (811/712)	Apply CC-CV charge: cutoff voltage at 4.25 V, stop at 0.05C current	Record and store charge curve with timestamp
4	Discharge 0% SOC under 1C or 1/3C (811/712)	CC discharge to cutoff voltage	Record and store V and T every 5 min
5	Set temp to thermal equilibrium	Set chamber temp to 25 °C/45 °C, hold 30 min after reaching T_0_ to ensure battery T_core = T_ambient	Equilibrium condition: ΔT_batt ≤ ±0.3 °C/10 min
6	Initiate 30 min idle period	Rest battery (no load)	Record OCV every 10 min
7	Charge to 100% SOC under 1C or 1/3C (811/712)	Same as Step 3, record full capacity after charge	Compute fade rate vs. initial capacity
8	Discharge 0% SOC under 1C or 1/3C (811/712)	Same as Step 4, record discharged capacity after discharge	Compute fade rate vs. initial capacity
9	Repeat steps 7–8/each 100 cycles	Every 100 cycles, record total capacity fade (%)	Stop test if swelling/leakage occurs
10	Repeat steps 2–9	If capacity < 80% stop test and change cell	Recalibrate parameters

**Table 3 micromachines-16-01075-t003:** Comparison of fading rate of battery cells with various Ni content NMC cathodes.

Cathode	Manufacturer	Voltage (V)	Fading Rate (%/Cycle)	Temperature (°C)/Current Rate (C)	Calculated Duration (Cycle)	Ref.
NMC111	Self-synthesized	4.4	0.0204	25/1	1–300	[25]
NMC532	LiFun Technology (Zhuzhou, China)	4.4	0.00522	40/0.33	1–1667	[43]
NMC622	Self-synthesized	4.2	0.0175	25/1	1–1000	[39]
NMC712	Farasis Energy (Ganzhou, China)	4.25	0.0191	45/0.33	1–600	This study
NMC811	Farasis Energy (Ganzhou, China)	4.25	0.0234	25/1	6–1080	This study
NMC89	Self-synthesized	4.2	0.0362	25/1	1–1000	[21]

## Data Availability

The original contributions presented in the study are included in the article, further inquiries can be directed to the corresponding author.

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
