# Peer review of "Effects of Ni Content on Energy Density, Capacity Fade and Heat Generation in Li[NixMnyCoz]O2/Graphite Lithium-Ion Batteries"

_micromachines, 2025, doi:10.3390/mi16101075_

Round 1

Reviewer 1 Report

Comments and Suggestions for Authors

This manuscript systematically investigates the influence of Ni content on the performance of NMC ternary cathode materials by combining experimental studies with simulations, addressing a problem of practical significance. The article is well-structured and supported by a relatively comprehensive dataset; only several points require adjustment.

1.Figure 10 appears to truncate a substantial portion of the data; the authors should ensure that the data are presented in their entirety.

2. In Table 3, it is recommended to express the fading rate in percentage terms, as the current representation is not intuitive.

3. Some fonts appear inconsistent, leaving traces that resemble AI-generated text, for example, the superscripts in Ni²⁺/Ni⁴⁺ redox couple.

4.   The COMSOL module employed should be specified. In the Experimental section, the governing equations used should be provided, including the form of the modified Butler–Volmer equation and the meaning of the corresponding parameters.

Reviewer 2 Report

Comments and Suggestions for Authors

The research presented here by the authors details the impact of Ni content in cathode on three major factors of batteries, viz., performance, capacity fading and heat evolution. The authors demonstrate that capacity of ternary materials are larger than LFP and more with higher Ni content, capacity fades faster with higher nickel content, relationship of heat generation with Ni content (all of these are known facts). All the mentioned factors and parameters that authors have demonstrated here are very well know to the community. No new learnings or concepts are seen throughout the manuscript. Authors should try to come back with some originality. So of the questions to improve the manuscript content are:

  1. Authors should provide more details and specifications about the cells used for the study like manufacturer, voltage, capacity, etc.
  2. How are the authors able to make statistically significant conclusions from just 2 cells?
  3. What was the reason to exclude NMC111?
  4. What would have been the observations for LiNiO2?
  5. Figure 2 seems to have been jumbled up due to Word to PDF conversion. Please fix it.
  6. Does electrolyte or anode have any influence on the major conclusions that authors have drawn from the research?
  7. What is the anode material had Si, then would it impact the observations?
  8. What additives were added to the electrolyte?
  9. How much loss was observed during the formation cycles?
  10. Would the observations be different if the batteries were reserve lithium-ion batteries (RLIBs) or LMBs?

Round 2

Reviewer 2 Report

Comments and Suggestions for Authors

Thank you for answering the questions asked.